# Gene regulation by a protein translation factor at the single-cell level

**Roswitha Dolcemascolo**[ID][☯]**, Lucas Goiriz**[ID][☯]**, Roser Montagud-Martínez**[ID]**, Guillermo Rodrigo**[ID]*

Institute for Integrative Systems Biology (I2SysBio), CSIC–University of Valencia, Paterna, Spain

☯ These authors contributed equally to this work.
* guillermo.rodrigo@csic.es

## Abstract

Gene expression is inherently stochastic and pervasively regulated. While substantial work combining theory and experiments has been carried out to study how noise propagates through transcriptional regulations, the stochastic behavior of genes regulated at the level of translation is poorly understood. Here, we engineered a synthetic genetic system in which a target gene is down-regulated by a protein translation factor, which in turn is regulated transcriptionally. By monitoring both the expression of the regulator and the regulated gene at the single-cell level, we quantified the stochasticity of the system. We found that with a protein translation factor a tight repression can be achieved in single cells, noise propagation from gene to gene is buffered, and the regulated gene is sensitive in a nonlinear way to global perturbations in translation. A suitable mathematical model was instrumental to predict the transfer functions of the system. We also showed that a Gamma distribution parameterized with mesoscopic parameters, such as the mean expression and coefficient of variation, provides a deep analytical explanation about the system, displaying enough versatility to capture the cell-to-cell variability in genes regulated both transcriptionally and translationally. Overall, these results contribute to enlarge our understanding on stochastic gene expression, at the same time they provide design principles for synthetic biology.

## Author summary

In the cell, proteins can bind to DNA to regulate transcription as well as to RNA to regulate translation. However, cells have mainly evolved to exploit transcription factors as specific gene regulators, while translation factors have remained as global modulators of expression. Consequently, transcription regulation has attracted much attention over the last years to unveil design principles of genetic organization and to engineer synthetic circuits for cell reprogramming. In this work, the phage MS2 coat protein was exploited to regulate the expression of a green fluorescent protein at the level of translation. This synthetic system was instrumental to gain fundamental knowledge on stochasticity and regulation at an overlooked level within the genetic information flow.

**Data Availability Statement:** All data are in the manuscript and/or supporting information files.

**Funding:** This work was supported by the grants H2020-MSCA-ITN-2018 #813239 (RNAct) from the European Commission and PGC2018-101410-

B-I00 (SYSY-RNA) from the Spanish Ministry of Science, Innovation, and Universities (co-financed by the European Regional Development Fund) to GR. RD acknowledges a Marie Curie fellowship linked to MSCA-ITN-2018 #813239 and LG a CSIC JAE Intro fellowship (Consejo Superior de Investigaciones Científicas). The funders had no role in study design, data collection and analysis, decision to publish, or preparation of the manuscript.

**Competing interests:** The authors have declared that no competing interests exist.

## Introduction

The ability to map a given genotype to its corresponding phenotype is perhaps the biggest pursuit in molecular biology [1], especially in the post-genomic era, as it can provide fundamental insight and predictive power on natural evolution, with clear applications in biomedicine and ecology. However, it is well established that the very same genotype can lead to phenotypic heterogeneity in a non-changing environment [2]. This is the consequence of the inherent stochasticity of the different biochemical reactions that are needed for gene expression [3]. While stochastic events are often seen as undesirable, as they are when the optimal gene expression levels are lost [4], we now realize that a noisy gene expression can also be advantageous for the cell population to face environmental changes or induce time-dependent behaviors [5]. In this regard, substantial progress has been made over the last years to quantitatively understand and model this non-genetic variability (noise). However, there are still numerous questions regarding the mechanisms that produce and regulate noise in gene expression.

Motivated by the prevalence of transcriptional regulations in the cell [6], previous work focused on studying the emergence and propagation of noise in genes regulated transcriptionally [3,7]. For example, we now appreciate that some promoters can generate bursts of expression as a consequence of a stochastic switching in their activity [8], that the sign of the regulation determines the best way to extract information from the environment [9], and that the stochastic fluctuations can inform about the underlying regulation when time is considered [10]. In addition, recent studies also focused on post-transcriptional regulations implemented by small non-coding RNAs (in particular, by microRNAs in eukaryotic cells) [11,12]. These studies concluded that microRNAs, by controlling the messenger RNA (mRNA) abundance, can suppress part of the noise generated at the level of transcription, hence resulting in ideal genetic elements to engineer robust circuits. Nevertheless, studies on cell-to-cell variability when protein expression is regulated at the level of translation are scarce, especially when the regulation is exerted by a translation factor. We just know that structured 5' untranslated regions (UTRs) can generate noise in protein expression [13], which can even be tuned by *trans*-acting small RNAs [14].

The importance of studying how stochastic gene expression is generated and regulated at different levels in the genetic information flow lies in the fact that living cells implement highly intricate circuitries for multiple signal integration that allow displaying a variety of phenotypes. Certainly, this signal integration becomes easier and more scalable if different layers are exploited (*e.g.*, transcriptional, translational, and post-translational), and this is precisely what has evolved in nature. Only by understanding the particularities of each regulatory mode, we can rationalize the impact of gene expression on the cell behavior. As highlighted before, more studies on stochastic gene expression regulated at a layer other than transcription are mandatory, especially because there are important phenotypes in nature that arise as a consequence of changing expression translationally.

In this work, we exploited the bacteriophage MS2 coat protein (MS2CP) as a translation factor [15] to engineer a basic synthetic regulatory circuit from which to study stochastic gene expression when it is regulated translationally. In the natural context, in addition to be a structural protein to form the virion, MS2CP blocks the translation of the viral replicase upon binding to an RNA hairpin in the corresponding 5' UTR that contains the ribosome binding site (RBS) and the start codon [16]. Over the years, MS2CP has been used for many applications due to its strong binding affinity to RNA, such as the subcellular tracking of mRNAs with time and space [17], the study of protein-RNA interactions *in vivo* [18], the development of CRISPR scaffold RNAs for programmable transcription regulation (CRISPR stands for clustered regularly interspaced short palindromic repeats) [19], and the construction of nanoscale architectures that can serve, for example, to improve enzymatic reactions [20]. With our engineered

circuit, we examined gene expression in single cells by using a double reporter system to monitor both the regulator (MS2CP) and the regulated gene, and we also developed a mathematical model to provide a predictive quantitative foundation of the system.

## Results

### Regulation of translation with an RNA-binding protein in single cells

We engineered a synthetic genetic system in *E. coli* in which the RNA-binding protein MS2CP acts as a protein translation factor (**Fig 1A**). MS2CP was expressed from a synthetic PL-based promoter repressed by LacI (named as PLlac) [21] in a medium copy number plasmid (about 80 copies/cell). This allowed controlling the expression of the regulator (at the transcriptional level) with isopropyl β-D-1-thiogalactopyranoside (IPTG). In addition, we fused the enhanced blue fluorescent protein 2 (eBFP2) [22] to the N terminus of MS2CP (leading to eBFP2-MS2CP) in order to monitor its expression. As a regulated element, here we used the superfolder green fluorescent protein (sfGFP) [23], which was expressed from a constitutive promoter in a low copy number plasmid (about 15 copies/cell). The wild-type RNA motif recognized by MS2CP (with a dissociation constant of about 3 nM) [24] was placed in frame just after the start codon of sfGFP. In this way, MS2CP can block the progression of the ribosome on the regulated gene in the initial phase [15]. This mode of action differs from the natural one, in which MS2CP prevents translation initiation rather than elongation [16]. The resulting circuit behaves like an inverter considering IPTG as input and sfGFP as output, MS2CP being an internal regulator that operates at the level of translation.

We performed single-cell measurements of blue and green fluorescence by flow cytometry for a concentration gradient of IPTG (9 conditions) in order to quantitatively study the stochastic regulatory dynamics of this engineered system (**Fig 1B** and **1C**). We found a substantial down-regulation of sfGFP (about 50-fold in expression) as a consequence of the action of MS2CP on the cognate mRNA. From these data, we calculated the mean and the noise of expression for both eBFP2-MS2CP and sfGFP (the noise as the square of the coefficient of variation) [3], which were represented as a function of IPTG (**Fig 1D** and **1E**). The mean gives the average position of the population, and the noise is a measure of the cell-to-cell variability. These measurements were repeated for different populations, finding consistency in the results (**S1 Fig**). We then constructed a mathematical model relying on a series of algebraic equations from basics on the biochemistry of gene expression and molecular noise propagation [7]. To derive these mathematical expressions for the mean and the noise, we constructed a system of stochastic differential equations for mRNA and protein expression following the Langevin formalism. The rates of concentration changes were subject to stochastic fluctuations of intrinsic and extrinsic nature. This system was analytically solved in steady state with the mean-field approximation for the fluctuations. With a suitable parameterization, our model was able to recapitulate with reasonable agreement the values of mean expression and noise for both eBFP2-MS2CP and sfGFP, highlighting the functional form of the different dose-response curves. In particular, the mean expressions follow Hill-Langmuir equations and the noises non-monotonous curves presenting a maximum at an intermediate IPTG concentration. Indeed, the peak-like noise curve is a consequence of a sigmoidal dynamics at the population level. We also observed that the noise levels in sfGFP are lower than in eBFP2 for all IPTG concentrations. We also performed numerical simulations of the stochastic differential equations (**S2 Fig**), finding good agreement with the analytical results, as well as sensitivity analyses to reveal the effect of perturbations in the adjusted parameters (**S3 Fig**), highlighting how the curves of mean expression and noise shift in one direction and even change in form. In addition, we represented the noise *versus* the mean to show the stochastic expression scaling laws

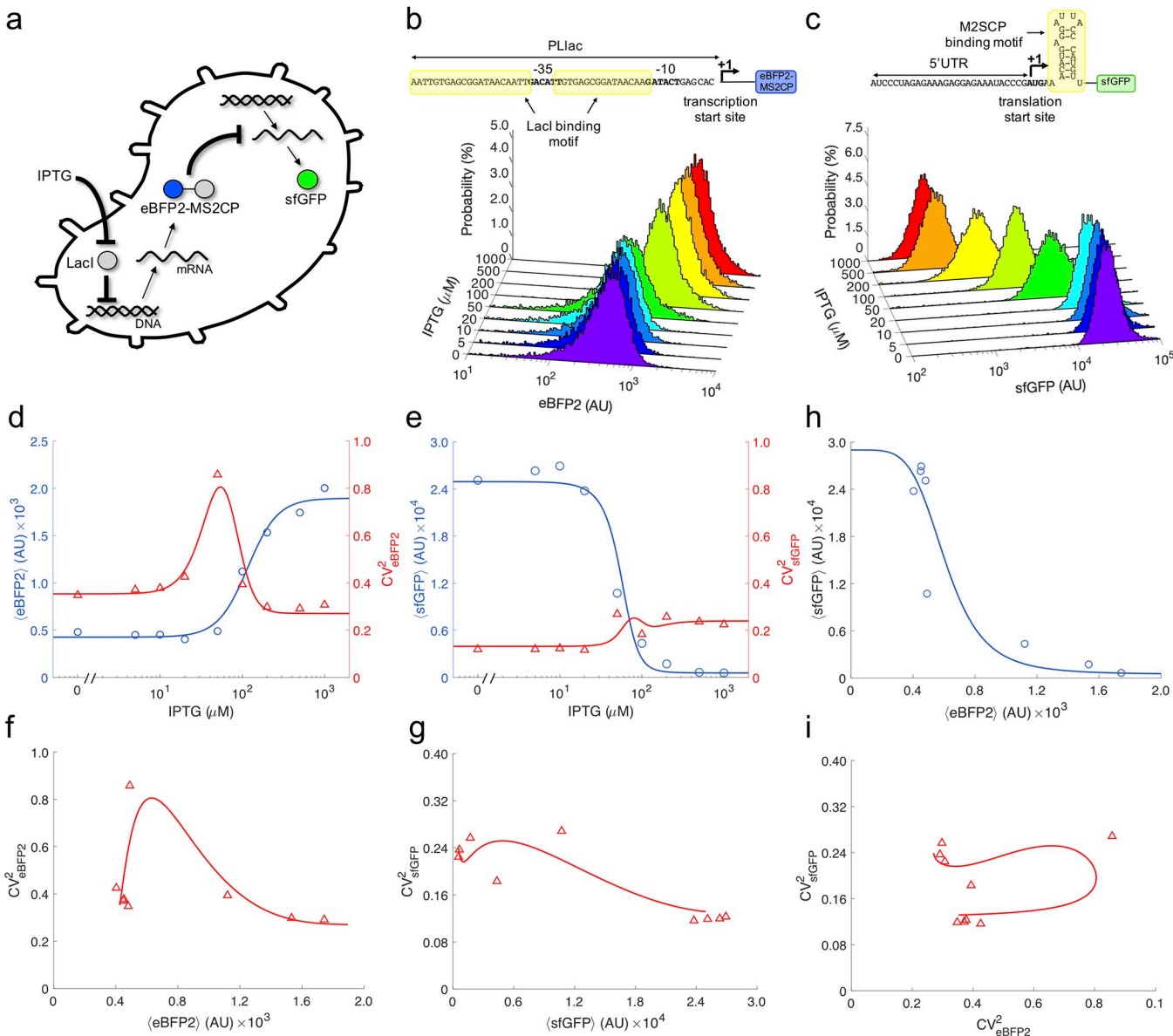

**Fig 1. Regulation with a protein translation factor.** a) Schematics of the gene regulatory system implemented in a bacterial cell. IPTG is the external molecule that controls the expression of the protein translation factor (eBFP2-MS2CP). sfGFP is the final output of the system. b) Histograms of single-cell fluorescence for eBFP2 (fused to the regulatory protein) for different induction conditions with IPTG. On top, sequence details of the *cis*-regulatory region (DNA level) for transcriptional regulation (PLlac promoter). c) Histograms of single-cell fluorescence for sfGFP (the regulated protein) for different induction conditions with IPTG. On top, sequence details of the *cis*-regulatory region (RNA level) for post-transcriptional regulation (MS2CP RNA motif). d) Mean and noise of expression for eBFP2 as a function of IPTG. e) Mean and noise of expression for sfGFP as a function of IPTG. f) Noise for eBFP2 as a function of mean expression. g) Noise for sfGFP as a function of mean expression. h) Transfer function of the post-transcriptional regulation in terms of mean expression. i) Transfer function of the post-transcriptional regulation in terms of noise. In plots d-i), points correspond to calculations from the experimental data, while solid lines to predictions with the mathematical model.

of the system (**Fig 1F and 1G**). The model was also explicative about the nonlinear transfer functions in terms of mean expression regulation (**Fig 1H**) and noise propagation (*i.e.*, how the noise of eBFP2-MS2CP impacts on the noise of sfGFP; **Fig 1I**). Together, these results indicated that the protein translation factor is a suitable element to control expression and that the cell-to-cell variability emerged at this level can be predicted with certain accuracy.

## Noise analysis in transcription and translation regulation

To further analyze the stochastic behavior, we looked inside the noise. That is, we inspected how a particular noise level is achieved. For that, we first decomposed the total noise of both eBFP2-MS2CP and sfGFP into three fundamental components: extrinsic noise, intrinsic noise, and regulation noise. Extrinsic noise comes from replication and variability in the cellular machinery, intrinsic noise is a consequence of a low number of molecules, and regulation noise accounts for the noise that is propagated from the regulator to the regulated gene [3,7,25]. In previous work, the regulation noise has been considered as a part of the extrinsic noise. Here, we separate this component to study more in detail the stochastic gene expression when it is regulated. Assuming independence between the different stochastic sources, we were able to end with compact mathematical expressions for the noise in which the different components were identified, although at the cost of introducing some inaccuracies since the extrinsic noise may correlate responses. Along the IPTG gradient and according to our mathematical model, the extrinsic noise of the system is constant, the intrinsic noise decreases in the case of eBFP2-MS2CP and increases in the case of sfGFP (as this noise scales inversely with the expression level), and the regulation noise follows a peak-like curve (**Fig 2A–2H**). Even though for both eBFP2-MS2CP and sfGFP the functional form of the regulation noise is similar, peaking at 50–75 μM IPTG, the maximal noise level is much lower (about four times) in the case of sfGFP. This suggested that with a translational control the noise of the regulator is buffered, *i. e.*, the fluctuations of MS2CP expression are manifested on sfGFP expression in lower extent than the fluctuations of LacI expression on MS2CP expression. This is because in a scenario of translational control the regulated gene is constantly transcribed at high levels, where fluctuations in the number of mRNAs per cell are small in comparison with the mean quantity that is produced. Thus, the transcriptional noise is not significant. In addition, the regulation enters at the level of translation, which prevents the typical amplification process of the noise of the regulator that occurs with a transcriptional control [26]. Indeed, in such a scenario, the transcription rate can be quite low when the promoter is repressed, thereby leading to substantial fluctuations in mRNA amount in comparison with the mean production. Furthermore, in the post-transcriptional case, the fluctuations in mRNA abundance can partly be absorbed by the effect of the translation factor, controlling the number of mRNAs available for translation. This has already been discussed in the case of regulatory RNAs [27], but it also applies to the case of a protein translation factor. Consequently, we can argue that the noise in the regulated gene is reduced when the regulation occurs at the level of translation.

In addition, we aimed at predicting the shape of the whole distribution of protein expression and not only the particular noise value. To this end, we considered a Gamma distribution, which has been shown to describe quite well the stochasticity of genetic systems [28] and which emerges from *ab initio* calculations [29]. The distribution of protein expression is instrumental to appreciate the degree of heterogeneity in the production with time and from cell to cell (assuming ergodicity). Here, by defining the Gamma shape parameter as the inverse of the noise (equal to the mean square divided by the variance) and the Gamma scale parameter as the product between the mean and the noise (*i.e.*, an effective Fano factor), we were able to predict the distributions for both eBFP2-MS2CP and sfGFP (**Fig 2I–2L**). This was done with the values of mean expression and noise given by the mathematical model. As a result of a transcriptional control, the Gamma scale parameter for eBFP2-MS2CP depends on the translation rate; but in the case of sfGFP, as this element is controlled at the level of translation, the Gamma scale parameter is nearly independent of that rate. Importantly, these theoretical distributions were very close to those fitted directly against the experimental data (**S4 Fig**), although some discrepancies were observed between the data and the model at intermediate

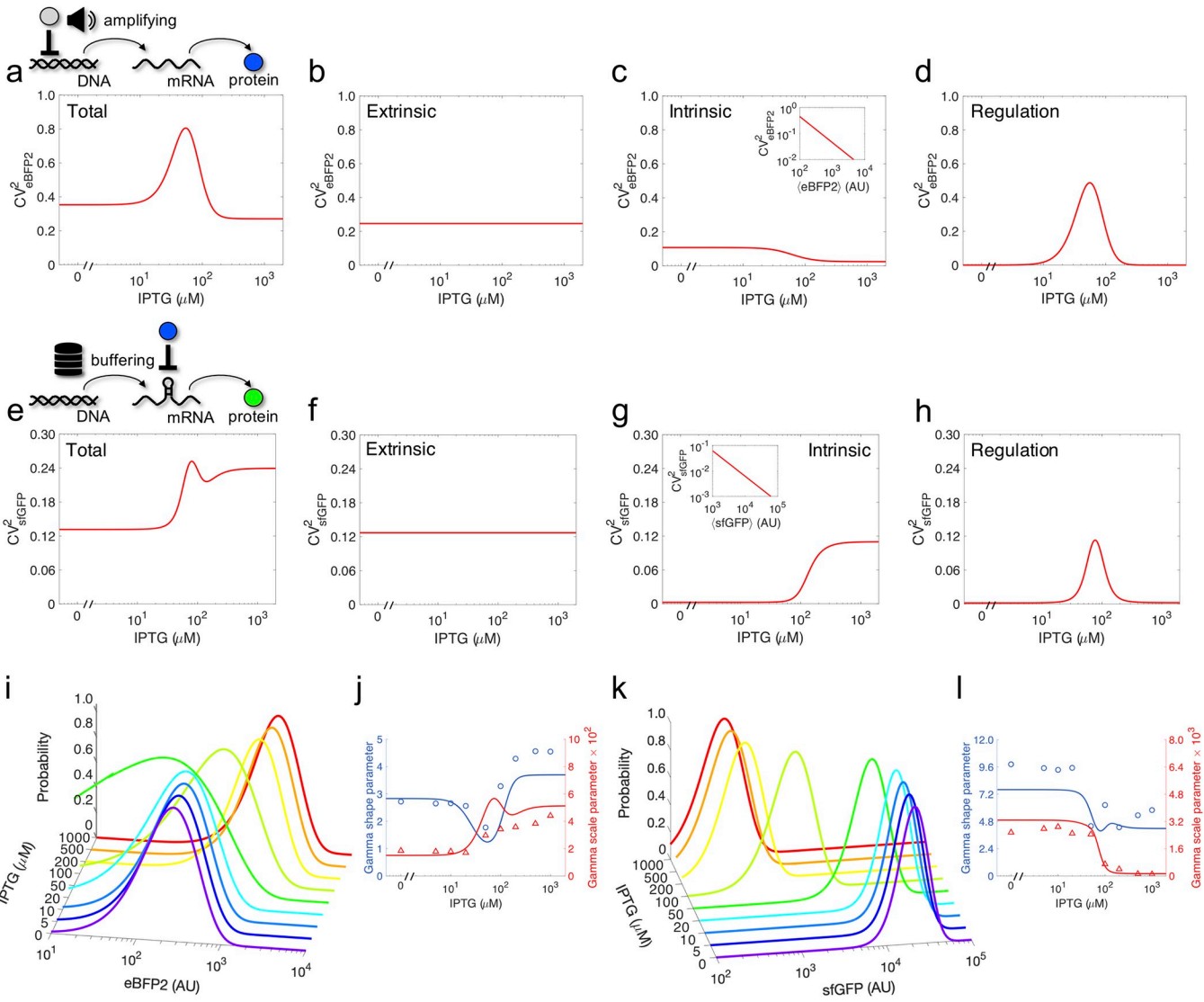

**Fig 2. Detailed analysis of stochastic gene expression.** a) Model-based calculation of the total noise in eBFP2 expression as a function of IPTG. On top, schematics of the amplification effect by the transcriptional regulation. b-d) Decomposition of the total eBFP2 noise into extrinsic, intrinsic, and regulation noise components. e) Model-based calculation of the total noise in sfGFP expression as a function of IPTG. On top, schematics of the buffering effect by the post-transcriptional regulation. f-h) Decomposition of the total sfGFP noise into extrinsic, intrinsic, and regulation noise components. Insets in c,g) show the scaling of the intrinsic noise with the mean expression. i) Predicted eBFP2 fluorescence distributions for different induction conditions with IPTG (Gamma distributions). j) Gamma shape and scale parameters for eBFP2. k) Predicted sfGFP fluorescence distributions for different induction conditions with IPTG (Gamma distributions). l) Gamma shape and scale parameters for sfGFP. In plots j,l), points correspond to calculations from the experimental values of mean and noise, while solid lines to predictions with the mathematical model.

IPTG concentrations. Overall, this highlighted the generality of the Gamma distribution to describe genetic systems regulated at both transcriptional and translational levels.

## Examination of global effects on regulated gene expression

Subsequently, we decided to study how global perturbations can impact the single-cell response of the system. To this end, we considered the effect of a global signal affecting translation. Here, we used sublethal concentrations of tetracycline (TC), a bacteriostatic antibiotic known to inhibit the formation of active ribosomes (**Fig 3A**) [30]. Paradoxically, this inhibition leads to an increase in translation rate as a result of an over-production of ribosomes (a

global response mechanism in bacteria against this type of antibiotics) [31,32]. That is, the cell is able to sense that a substantial amount of ribosomes is being inhibited upon binding to TC and produces more. In particular, TC binds to the 30S subunit and interferes with the transfer RNAs (tRNAs). In turn, the cell growth rate is compromised due to the action of TC. Importantly, this parameter has been shown to modulate the mean and noise of gene expression [33,34], so we decided to exploit it as a predictor variable. Over a two-dimensional concentration gradient of IPTG and TC (81 conditions), we first generated growth curves (**S5 Fig**). Basically, only TC showed a significant impact on growth rate (**Fig 3B**), with a maximal reduction of almost 3-fold, which was well explained by a Michaelis-Menten function (**Fig 3C**).

In parallel, we performed single-cell measurements of blue and green fluorescence for each condition (**Fig 3D**). We observed that the mean expression levels of both eBFP2-MS2CP and sfGFP remained almost constant at low TC concentrations, but they increased significantly from 500 ng/mL TC, irrespective of the induction with IPTG (**Fig 3E** and **3F**). Because protein expression comes from the ratio between the protein synthesis rate (accounting for both transcription and translation) and the growth rate (in the case of stable proteins, as it is the case here), this indicated that the protein synthesis rate of both eBFP2-MS2CP and sfGFP scales with the growth rate (**Fig 3G** and **3H**). It was interesting to note here the logical NOR behavior of the sfGFP synthesis rate and the difference between protein expression and synthesis rate. In addition, we calculated the noise levels for each condition (**Fig 3I** and **3J**). We observed that the regulation noise decreases with TC for both eBFP2-MS2CP and sfGFP, as well as that TC leads to a substantial increase in the sfGFP noise when this gene is fully repressed by MS2CP.

## Integrative modeling of the deterministic and stochastic dynamics

Importantly, we noted that the cell volume increases as a consequence of TC (**Fig 4A**; see also **S6 Fig** where we show an exponential trend), which agrees with previous observations [35]. This entails the necessity of a higher number of MS2CP molecules per cell to repress the target gene. Moreover, it is known that the number of total ribosomes decreases linearly with the growth rate when it is modulated by an antibiotic (*i.e.*, the slower the replication, the larger the number of ribosomes) [31]. This leads to a reciprocal function with the growth rate to describe the translation rate of a given gene [32]. That is, the translation rate scales inversely with the growth rate. It is also known that the transcription rate (mRNA production) scales linearly with the growth rate [33]. In terms of number of proteins per cell, this effect is cancelled out by the effective dilution due to cell division. By introducing into our mathematical model these dependencies, we were able to predict with relatively good agreement the impact of TC on mean expression for both eBFP2-MS2CP and sfGFP (**Fig 4B** and **4C**).

To further comprehend the interplay between gene regulation and cell growth, we used the model to predict the protein synthesis rate. Remarkably, we found that for eBFP2-MS2CP the same curve is explicative for all induction conditions with IPTG, provided the values are relativized to the case of no TC (**Fig 4D**). This is because eBFP2-MS2CP is regulated transcriptionally, and in turn the transcription factor of the system (LacI) is expressed constitutively and modulated by IPTG in terms of activity at the post-translational level. Hence, there is a decoupling between the regulation and the effect of growth rate on expression. A minimum in eBFP2-MS2CP synthesis rate was found at a growth rate of about 0.45 h$^{-1}$, which comes from the fact that the protein synthesis rate is modeled by a rational function with the growth rate. In essence, at low TC concentrations, the transcription rate is reduced, but the translation rate remains almost constant. However, at high TC concentrations, the translation rate very much increases and dominates over the transcription rate. By contrast, we found that the relative sfGFP synthesis rate strongly depends on IPTG and that this dependence is well captured by the model (**Fig 4E**). In this case,

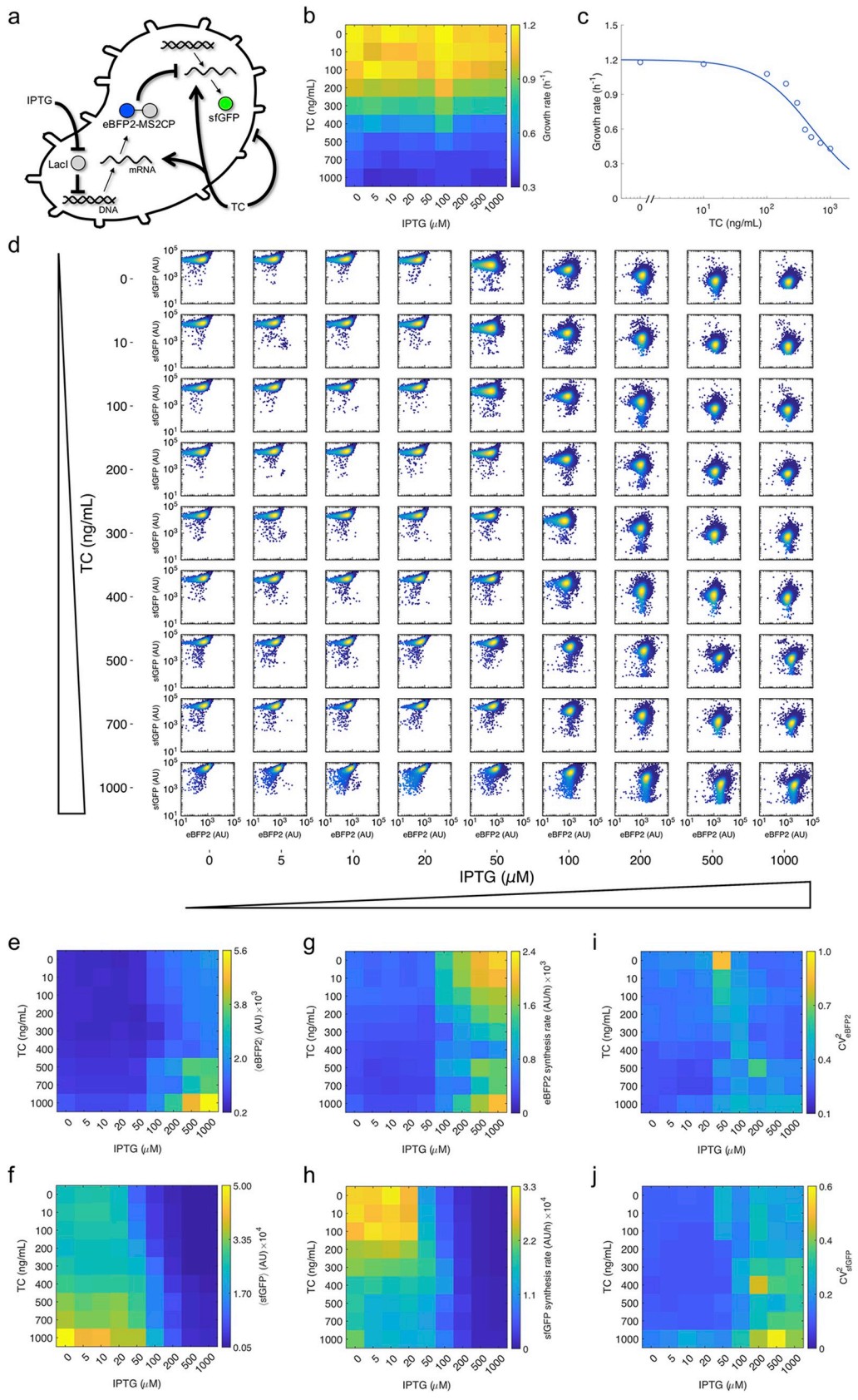

**Fig 3. Growth-dependent regulation with a protein translation factor.** a) Extended schematics of the gene regulatory system in which TC further modulates it through its negative impact on growth rate and positive impact on translation rate (global effects). b) Heatmap of the mean growth rate as a function of IPTG and TC. c) Dose-response curve between growth rate and TC. Points correspond to experimental data, while solid line comes from the mathematical model. d) Projected two-dimensional histograms of single-cell fluorescence for eBFP2 and sfGFP for different induction conditions with IPTG and TC. e) Heatmap of the mean eBFP2 fluorescence as a function of IPTG and TC. f) Heatmap of the mean sfGFP fluorescence as a function of IPTG and TC. g) Heatmap of the mean eBFP2 synthesis rate as a function of IPTG and TC. h) Heatmap of the mean sfGFP synthesis rate as a function of IPTG and TC. i) Heatmap of the eBFP2 noise as a function of IPTG and TC. j) Heatmap of the sfGFP noise as a function of IPTG and TC.

the number of MS2CP molecules per cell changes with IPTG, so there is a coupling between the translational regulation and the effect of growth rate on expression. While at low IPTG concentrations the relative sfGFP synthesis rate follows the aforementioned trend for eBFP2-MS2CP, at high IPTG concentrations there is a maximum at a growth rate of about 0.65 h$^{-1}$ (it was particularly pronounced at the intermediate value of 100 μM). **Fig 4F** illustrates how the transfer function in terms of mean expression varies with the TC concentration (*i.e.*, by increasing the maximal expression level and shifting the inflexion point towards the right).

Finally, we applied the model to project the noise in protein expression. In this case, we needed to introduce a phenomenological dependence with the growth rate on three noise-related parameters to explain the data. In particular, we set that the noise in LacI expression scales with the square of the growth rate (*i.e.*, LacI expression varies from cell to cell in greater extent when cells grow faster) and that the extrinsic noise of both eBFP2-MS2CP and sfGFP scales with the inverse of the growth rate by following the translation rate (*i.e.*, the extrinsic noise is higher at lower growth rates, which seems in tune with recent experiments characterizing genome-wide noisy expression levels [36]). While for eBFP2-MS2CP the noise decreases or remains constant with the growth rate (**Fig 4G**), for sfGFP the noise presents a more complex trend (almost constant at low IPTG concentrations and with a maximum at high IPTG concentrations; **Fig 4H**). In turn, **Fig 4I** illustrates how the transfer function in terms of noise varies with the TC concentration, showing how the belly shape is reduced with TC, which indicates that noise propagation through the translation factor is less significant (*i.e.*, it is masked) when the growth rate is low. That is, the regulation noise term, which quantifies how much fluctuation sfGFP perceives from MS2CP, becomes smaller than the other noise terms (intrinsic and extrinsic) with TC. Arguably, at high growth rates, when global perturbations are small, a translation factor is superior to a transcription factor because it is able to regulate gene expression without transmitting much noise. However, this is not the case at low growth rates, when global perturbations become substantial, due to a poor signal-to-noise ratio. Together, these results highlight the complex impact of growth rate in the system and serve to appreciate how global and local regulatory mechanisms interplay in the cell.

## Discussion

Our development follows previous work on exploiting RNA-binding proteins as translation factors to engineer gene circuits [15]. In this work, we focused on quantitatively studying the stochastic behavior of this type of circuits (*i.e.*, noise generation and propagation in gene expression when it is regulated at the level of translation). Our results show that the protein-RNA interaction in this case leads to a significant down-regulation in expression of about 50-fold by blocking the progression of the ribosome on the target mRNA, which is comparable to transcriptional fold-changes. In addition, a general mathematical framework was shown suitable to describe the stochastic behavior in regulations exerted by both transcription and translation factors. Noise propagation from gene to gene is buffered when the regulator acts at the level of translation, as the amplification process by transcription is avoided, and a Gamma distribution properly

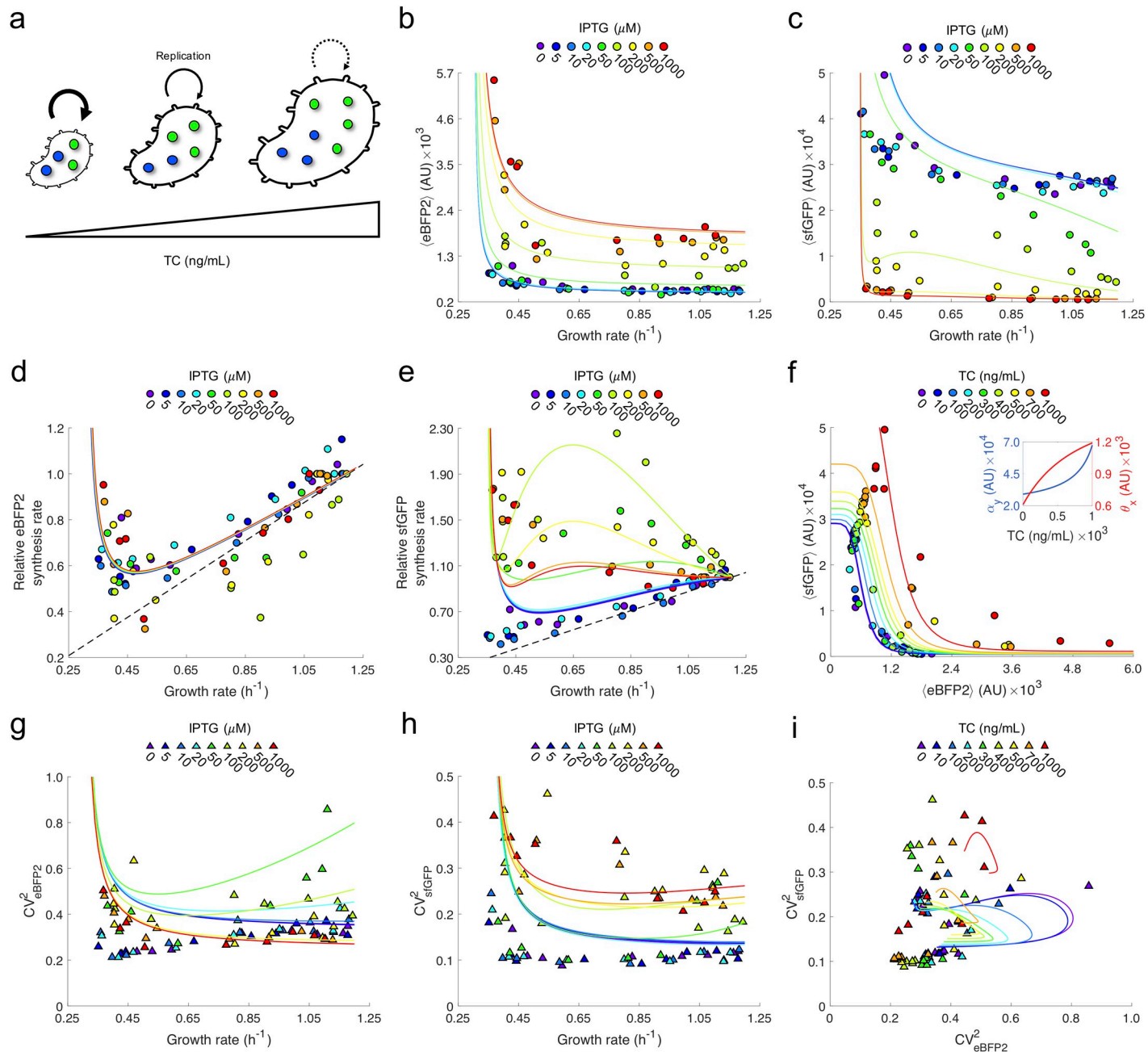

**Fig 4. Detailed analysis of stochastic gene expression modulated by growth rate.** a) As TC increases, cells grow slower and bigger. b) Mean eBFP2 expression as a function of growth rate for each IPTG condition. c) Mean sfGFP expression as a function of growth rate for each IPTG condition. d) Relative mean eBFP2 synthesis rate as a function of growth rate for each IPTG condition. e) Relative mean sfGFP synthesis rate as a function of growth rate for each IPTG condition. In d,e), the values are relative to the case TC = 0, and the dashed line corresponds to a linear dependence (which comes from a null model in which the translation rate is not affected by the growth rate). f) Transfer function of the post-transcriptional regulation in terms of mean expression for each TC condition. The inset shows the effect of TC on the sfGFP mRNA synthesis rate (included in $\alpha_y$, as the growth rate decreases) and the effective dissociation constant between eBFP2-MS2CP and sfGFP mRNA ($\theta_x$, as the volume increases) according to the mathematical model. g) eBFP2 noise as a function of growth rate for each IPTG condition. h) sfGFP noise as a function of growth rate for each IPTG condition. i) Transfer function of the post-transcriptional regulation in terms of noise for each TC condition. In plots b-i), points correspond to calculations from the experimental data, while solid lines to predictions with the mathematical model.

parameterized can provide deep analytical explanations about the resulting cell-to-cell variability [28]. By modulating the cellular growth rate, we also reported an interplay between global and local regulatory mechanisms in the cell that affect both the mean expression and noise levels.

It is important to notice that the growth rate that we measured here corresponds to an average of the population (as we calculate it having monitored absorbance with time for a culture). Nevertheless, each cell grows differently, especially when the culture is under the effect of TC. In this regard, a mathematical model incorporating such heterogeneous growth might explain better the observed noise patterns [37]. Another limitation of our work is the assumption that the noise sources are independent, which allowed us to derive a compact mathematical expression for the noise of sfGFP. This is not strictly true since eBFP2-MS2CP and sfGFP share extrinsic noise sources [7]. Yet, we expect reliability in the conclusions derived from this study.

Here, we exploited the viral protein MS2CP to implement the regulatory system, but in principle other RNA-binding proteins might be used. For example, the bacteriophage PP7 coat protein or the *Mycobacterium* enzyme PyrR [16] are suitable elements from which to engineer orthogonal systems. In fact, given the plethora of RNA-binding proteins in nature, especially in eukaryotes [38], and noting that the regulatory mechanism only requires a tight protein-RNA interaction to interfere with the ribosome, multiple implementations might be achieved. Our mathematical model is general enough to describe these eventual implementations. We only expect to change the kinetic parameters for each particular protein, preserving the functional form. In principle, each RNA motif will lead to a different translation rate. In this regard, the predictability of the system might be strengthened by using the RBS calculator [39]. We also anticipate that the use of tandem repeats of the RNA motif might enhance the regulatory fold-change of the system. Furthermore, since RNA is a very versatile molecule, RNA-binding proteins can regulate gene expression through a variety of mechanisms acting post-transcriptionally, including the regulation of translation initiation, translation elongation, transcription termination, and RNA stability [40]. In prokaryotes, the regulation of translation initiation by controlling RBS accessibility is a widespread mechanism, but in eukaryotes the blockage of translation elongation has been observed in the case of Argonaute proteins [41]. Arguably, a blockage in the initial phase by MS2CP is key in our synthetic system. Further work should analyze how those other mechanisms generate and propagate noise.

In sum, our work provides new quantitative insights about the stochastic behavior in genes regulated translationally by RNA-binding proteins. Protein translation factors can integrate some advantages distinctively attributed to proteins (as transcription factors), such as the ability to transduce small signals and achieve large dynamic ranges, or to small RNAs, such as the ability to produce rapid responses and buffer transcriptional noise [27]. Furthermore, our work paves the way for engineering gene regulatory circuits with greater integrability and then sophistication. Certainly, the combination of different layers within the genetic information flow (*i.e.*, transcription and translation) leads to an easier integration of signals to achieve a given function [42]. Therefore, we envision that RNA-binding proteins will be of great utility in synthetic biology in the close future to face biotechnological and biomedical challenges.

## Materials and methods

### Strains, plasmids, and reagents

*E. coli* Dh5α was used for cloning purposes by following standard procedures. To express our genetic circuit, *E. coli* MG1655-Z1 (*lacI*+ and *tetR*+; kindly gifted by M.B. Elowitz) was used. This strain was co-transformed by electroporation with two plasmids, called pRKFR2 (kanR, pSC101-E93G ori) and pREP3 (camR, p15A ori; kindly gifted by J.A. Daròs). pRKFR2 contains the gene coding for MS2CP translationally fused to eBFP2 in its N terminus (eBFP2-MS2CP) under the transcriptional control of the inducible promoter PLlac. pREP3 contains the gene coding for sfGFP under the transcriptional control of the constitutive promoter J23119 and

the translational control of the MS2CP-recognizing RNA motif. The genetic cassettes were synthesized by IDT. LB medium was used for both overnight and characterization cultures. Kanamycin and chloramphenicol were used at the concentration of 50 μg/mL and 34 μg/mL, respectively. IPTG and TC were used as inducers of the system. The concentration gradient of IPTG that we tested was 0, 5, 10, 20, 50, 100, 200, 500, and 1000 μM, and the concentration gradient of TC was 0, 10, 100, 200, 300, 400, 500, 700, and 1000 ng/mL. Compounds provided by Sigma.

Note that LacI is overexpressed in MG1655-Z1 to efficiently regulate the PLlac promoter in the pRKFR2 plasmid. This overexpression adds to the wild-type expression of LacI. In addition, TetR is expressed in *E. coli* MG1655-Z1 (although it does not play any regulatory role), so it will bind to TC when this inducer is used. The titration effect will be more relevant at low concentrations of TC, although overall this will only mean an effective TC concentration slightly lower. The maximal TC concentration used here reduced significantly the growth rate of the cells, suggesting a marginal effect of TetR in this case.

## Growth curves

Cultures (2 mL) inoculated from single colonies (three replicates) were grown overnight in LB medium at 37˚C and 200 rpm. Cultures were then diluted 1:100 in fresh LB medium (2 mL) and were grown for 3 h at the same conditions to reach exponential phase (OD$_{600}$ around 0.5). Cultures were then diluted 1:50 in fresh LB medium (200 μL) to load a microplate (96 wells, black, clear bottom; Corning) with appropriate concentrations of IPTG and TC. The microplate was then incubated for 10 h at 37˚C and 1,000 rpm in a PST-60HL plate shaker (Biosan). Absorbance (600 nm) was measured every hour in a Varioskan Lux fluorometer (Thermo). The growth rate was calculated as the slope between absorbance (in log scale) and time during the exponential phase. Data analysis performed with MATLAB (MathWorks) and Python.

## Flow cytometry

Cultures (2 mL) inoculated from single colonies (four replicates) were grown overnight in LB medium at 37˚C and 200 rpm. Cultures were then diluted 1:100 in fresh LB medium (2 mL) and were grown for 3 h at the same conditions to reach exponential phase (OD$_{600}$ around 0.5). Cultures were then diluted 1:50 in fresh LB medium (200 μL) to load a microplate (96 wells, black, clear bottom; Corning) with appropriate concentrations of IPTG and TC. The microplate was then incubated at 37˚C and 1,000 rpm in a PST-60HL plate shaker (Biosan) until cultures reached a sufficient OD$_{600}$ (a different incubation time for each TC concentration). Cultures (6 μL) were then diluted in PBS (1 mL). Fluorescence was measured in an LSRFortessa flow cytometer (BD); a 405 nm laser and a 450 nm filter for blue fluorescence, and a 488 nm laser and a 530 nm filter for green fluorescence. Events were gated by using the forward and side scatter signals and compensated (~$10^4$ events after this process). The mean value of the autofluorescence of the cells was subtracted in each channel to obtain a final estimate of expression (**S1 Data**). Data analysis performed with MATLAB and Python. The mean and the variance were calculated for each distribution after removing outliers, which served to compute the noise in gene expression.

## Deterministic mathematical modeling

We assumed that the cellular amount of regulatory protein (eBFP2-MS2CP) is proportional to the signal of blue fluorescence, and that the amount of regulated protein (sfGFP) is proportional to the signal of green fluorescence. Thus, with IPTG and TC being the two external

molecules of control, the Hill-Langmuir equations that dictate average protein expression (population measure) are

$$\langle \text{eBFP2} \rangle = \alpha_x \frac{\rho_x + \left(\frac{\text{IPTG}}{\theta_i}\right)^{n_i}}{1 + \left(\frac{\text{IPTG}}{\theta_i}\right)^{n_i}}$$

$$\langle \text{sfGFP} \rangle = \alpha_y \frac{1 + \rho_y \left(\frac{\langle \text{eBFP2} \rangle}{\theta_x}\right)^{n_x}}{1 + \left(\frac{\langle \text{eBFP2} \rangle}{\theta_x}\right)^{n_x}},$$

(1)

where $\alpha_x$ is the maximal protein level from the PLlac promoter (in presence of IPTG), $\rho_x$ the transcriptional repression fold by LacI, $\alpha_y$ the maximal protein level from the constitutive J23119 promoter, and $\rho_y$ the translational repression fold by eBFP2-MS2CP. In addition, $\theta_i$ is the effective dissociation constant between LacI and IPTG, $n_i$ the effective degree of cooperativity of LacI, $\theta_x$ the effective dissociation constant between eBFP2-MS2CP and the cognate RNA motif embedded within the sfGFP mRNA, and $n_x$ the effective degree of cooperativity of eBFP2-MS2CP. By using our data, the adjusted parameter values are $\alpha_x$ = 1,890 AU, $\rho_x$ = 0.225, $\theta_i$ = 116 μM, $n_i$ = 2.38, $\alpha_y$ = 29,000 AU, $\rho_y$ = 0.016, $\theta_x$ = 610 AU, and $n_x$ = 5 (upon varying IPTG, with no TC).

Moreover, if $\mu$ denotes the actual cell growth rate, which is modulated by TC and dictates the dilution rate of the proteins, it turns out that the following Michaelis-Menten equation

$$\mu = \frac{\mu_0}{1 + \frac{\text{TC}}{\theta_c}},$$

(2)

where $\mu_0$ is the maximal cell growth rate (in absence of TC) and $\theta_c$ the half maximal inhibitory concentration of TC. From the data, we obtained $\mu_0$ = 1.2 h$^{-1}$ and $\theta_c$ = 526 ng/mL (upon varying TC, with no IPTG).

To consider the impact of growth rate on protein expression, we noticed that protein expression is the ratio between the protein synthesis rate (SR) and the growth rate, and that the protein synthesis rate is the product between the mRNA amount and the translation rate ($\lambda_i$, for gene $i$). That is,

$$\langle \text{eBFP2} \rangle = \frac{\langle \text{SR}_{\text{eBFP2}} \rangle}{\mu} = \frac{\lambda_x \langle \text{mRNA}_{\text{eBFP2}} \rangle}{\mu}$$

$$\langle \text{sfGFP} \rangle = \frac{\langle \text{SR}_{\text{sfGFP}} \rangle}{\mu} = \frac{\lambda_y \langle \text{mRNA}_{\text{eBFP2}} \rangle}{\mu}.$$

(3)

Because it is known that the mRNA amount per cell is proportional to the growth rate (*i.e.*, the transcription rate increases as long as the cell grows faster, $\langle \text{mRNA} \rangle \propto \mu$) [33], the dependence of the protein synthesis rate on the growth rate is just given by the effect of the growth rate on the translation rate. Interestingly, the translation rate can be described by a Michaelis function of the number of active ribosomes [32], which is known to increase linearly when the growth rate decreases as a consequence of TC. We can then define two effective parameters ($\varepsilon_{1i}$ and

$\varepsilon_{2i}$, for gene $i$) to model such a dependence as $\lambda_i \propto \frac{\varepsilon_{1i}-\mu}{\varepsilon_{2i}-\mu}$ and then write for eBFP2 and sfGFP the following

$$\alpha_x(\mu) = \frac{\lambda_x(\mu)}{\lambda_x(\mu_0)}\alpha_x(\mu_0)$$
$$\alpha_y(\mu) = \frac{\lambda_y(\mu)}{\lambda_y(\mu_0)}\alpha_y(\mu_0).$$

(4)

Here, we adjusted $\varepsilon_{1x} = 0.196$ h$^{-1}$, $\varepsilon_{1x} = 0.299$ h$^{-1}$, $\varepsilon_{1y} = 0.246$ h$^{-1}$, and $\varepsilon_{1y} = 0.349$ h$^{-1}$.

In addition, we noticed that the cellular volume changes with the growth rate as a consequence of TC. By using the cube of the forward scatter signal (median of the population) as a proxy of the volume, a negative exponential trend was identified (*i.e.*, the volume increases with TC). Because the model considers as variables the amount of fluorescent proteins per cell and not the concentrations, the parameter $\theta_x$ needs to be corrected as

$$\theta_x(\mu) = e^{\delta(\mu_0-\mu)}\theta_x(\mu_0),$$

(5)

where here we took $\delta = 0.85$ h. In essence, when the volume increases, the number of proteins required to regulate the target gene are higher. Besides, the impact of growth rate on the transcriptional regulation was assumed negligible, as LacI is a highly expressed protein from the chromosome whose activity is modulated by a chemical inducer, which enters into the model in terms of concentration (*i.e.*, $\theta_i$ independent of $\mu$).

## Stochastic mathematical modeling

The expressions for the variances in gene expression (or fluorescence signal) can be derived by following some basic calculations (see **S1 Appendix** for a detailed calculation) [3,7]. We followed a Langevin formalism, which consists in introducing in the right-hand side of the differential equation a series of stochastic processes, and the mean-field approximation for the analytical treatment, which consists in assuming that the different fluctuation amplitudes depend on the deterministic solution. In brief, the variance for a given gene and induction condition can be decomposed into three different variances according to the nature of the molecular noise source. The variance coming from extrinsic noise can be assumed to scale with the square of the expression level (then leading to a constant term in gene expression noise), the variance from intrinsic noise with the expression level, and the variance from the regulatory protein noise with the square of the derivative of the transfer function (in terms of protein synthesis rate). In particular, we can write

$$\mathrm{CV}^2_{\mathrm{eBFP2}} = \underbrace{\eta_x^2}_{\text{extrinsic}} + \underbrace{\frac{\beta_x}{\langle\mathrm{eBFP2}\rangle}}_{\text{intrinsic}} + \underbrace{\frac{1}{2}\left(\frac{\alpha_x(1-\rho_x)\langle n_i\left(\frac{\mathrm{IPTG}}{\theta_i}\right)^{n_i-1}}{\theta_i\left(1+\left(\frac{\mathrm{IPTG}}{\theta_i}\right)^{n_i}\right)^2}\right)^2 \frac{\eta_{\mathrm{lac}}^2}{\langle\mathrm{eBFP2}\rangle^2}}_{\text{regulation}}$$

$$\mathrm{CV}^2_{\mathrm{sfGFP}} = \underbrace{\eta_y^2}_{\text{extrinsic}} + \underbrace{\frac{\beta_y}{\langle\mathrm{sfGFP}\rangle}}_{\text{intrinsic}} + \underbrace{\frac{1}{2}\left(\frac{\alpha_y(1-\rho_y)n_x\left(\frac{\langle\mathrm{eBFP2}\rangle}{\theta_x}\right)^{n_x-1}}{\theta_x\left(1+\left(\frac{\langle\mathrm{eBFP2}\rangle}{\theta_x}\right)^{n_x}\right)^2}\right)^2 \frac{\gamma_y\langle\mathrm{eBFP2}\rangle^2}{\langle\mathrm{sfGFP}\rangle^2}\mathrm{CV}^2_{\mathrm{eBFP2}}}_{\text{regulation}}, (6)$$

where $\eta_x^2$ and $\eta_y^2$ are two empirical constants that quantify the levels of noise of extrinsic nature

on eBFP2-MS2CP and sfGFP, respectively. Also, $\eta_{\text{lac}}^2$ is a constant that measures the noise in LacI expression, $\beta_x$ and $\beta_y$ the Fano factors of noise of intrinsic nature for eBFP2-MS2CP and sfGFP, respectively, and $\gamma_y$ a constant that accounts for the difference between fluorescence and number of molecules. Note that $\langle \text{eBFP2} \rangle = \langle \text{MS2CP} \rangle$ and $\text{CV}_{\text{eBFP2}}^2 = \text{CV}_{\text{MS2CP}}^2$. Note also that when a strong repression occurs at the level of translation the transcriptional noise can be neglected and then $\beta_y$ can be considered constant (*i.e.*, the Fano factor, in number of molecules per cell, can be approached by 1). By using our data, the adjusted parameter values are $\eta_x^2 = 0.246$, $\beta_x$ = 45.6 AU, $\eta_{\text{lac}}^2 = 6,470 \, \mu\text{M}^2$, $\eta_y^2 = 0.127$, $\beta_y$ = 61.9 AU, and $\gamma_y = 0.0233$ (upon varying IPTG, with no TC).

It is important to recall that the parameters $\alpha_x$, $\alpha_y$, and $\theta_x$ depend on the growth rate. Moreover, the intrinsic noise Fano factor for eBFP2-MS2CP is proportional to the translation rate ($\beta_x \propto \lambda_x$),

$$\beta_x(\mu) = \frac{\lambda_x(\mu)}{\lambda_x(\mu_0)} \beta_x(\mu_0), \tag{7}$$

but not the factor for sfGFP ($\beta_y$ nearly independent of $\lambda_y$), as sfGFP is regulated at the level of translation and then its mRNA is constitutively expressed. To explain our data, we introduced the following phenomenological expressions

$$\eta_{\text{lac}}^2(\mu) = \left(\frac{\mu}{\mu_0}\right)^2 \eta_{\text{lac}}^2(\mu_0)$$

$$\eta_x^2(\mu) = \frac{\lambda_x(\mu)}{\lambda_x(\mu_0)} \eta_x^2(\mu_0) \tag{8}$$

$$\eta_y^2(\mu) = \frac{\lambda_y(\mu)}{\lambda_y(\mu_0)} \eta_y^2(\mu_0).$$

In essence, this indicates that LacI expression varies from cell to cell in greater extent when cells grow faster, and that the extrinsic noise increases when the growth rate is very low.

Finally, we assumed that the stochastic gene expression follows a Gamma distribution (see **S2 Appendix** for a basic derivation) [28]. Then, the probability for a given expression level reads

$$P(\text{eBFP2}) = \frac{\text{eBFP2}^{a_x-1} e^{-\text{eBFP2}/b_x}}{\Gamma(a_x) b_x^{a_x}}$$

$$P(\text{sfGFP}) = \frac{\text{sfGFP}^{a_y-1} e^{-\text{sfGFP}/b_y}}{\Gamma(a_y) b_y^{a_y}}, \tag{9}$$

where $a_x$ and $a_y$ are the Gamma shape parameters for eBFP2-MS2CP and sfGFP, respectively, and $b_x$ and $b_y$ the Gamma scale parameters. Importantly, by knowing that for a Gamma distribution $ab$ is the mean and $ab^2$ the variance, these two parameters can be defined as

$$a_x = \frac{1}{\text{CV}_{\text{eBFP2}}^2}$$

$$a_y = \frac{1}{\text{CV}_{\text{sfGFP}}^2} \tag{10}$$

$$b_x = \langle \text{eBFP2} \rangle \text{CV}_{\text{eBFP2}}^2$$

$$b_y = \langle \text{sfGFP} \rangle \text{CV}_{\text{sfGFP}}^2.$$

This means that the Gamma shape parameter is directly the inverse of the noise, and that the

Gamma scale parameter depends on the translation rate in the case of eBFP2-MS2CP (transcription regulation) and is nearly independent of it in the case of sfGFP (translation regulation).

## Numerical simulations

The system of stochastic differential equations (see **S1 Appendix**) was solved numerically to obtain stochastic trajectories of mRNA and protein concentrations. For that, we followed an integration scheme previously described [43]. The colored stochastic processes (for extrinsic and regulation noise) were obtained from independent white stochastic processes. The system was solved in one time interval with the routine ode45s from MATLAB, considering constant the stochastic fluctuations in that interval. The values of the fluctuations were updated in each interval with the previous mRNA and protein concentrations. Negative concentration values were avoided.

## Supporting information

**S1 Fig. Reliability of the dose-response curve.** a) Mean of eBFP2 expression as a function of IPTG. b) Mean of sfGFP expression as a function of IPTG. c) Noise of eBFP2 expression as a function of IPTG. d) Noise of sfGFP expression as a function of IPTG. Points correspond to the values of the population shown in the main figures. Error bars correspond to standard errors calculated from four different populations. Solid lines correspond to predictions with the mathematical model.
(TIF)

**S2 Fig. Numerical simulations of stochastic dynamics.** a-d) Stochastic trajectories with time of eBFP2 and sfGFP for two different IPTG concentrations. In red, deterministic trajectories. The initial condition corresponds to the uninduced state in all cases. e-h) Histograms of protein expression computed from long trajectories. The Gamma distributions fitted against the experimental data (blue lines) were also represented.
(TIF)

**S3 Fig. Sensitivity analysis of the model parameters.** Plots of mean and noise of expression as a function of IPTG, where solid lines correspond to the dynamics predicted with the adjusted parameter, dotted lines to the dynamics if the parameter increases 2-fold, and dashed lines to the dynamics if the parameter decreases 2-fold.
(TIF)

**S4 Fig. Stochastic gene expression described by a Gamma distribution.** Histograms of experimental single-cell fluorescence for both a) eBFP2 and b) sfGFP for different induction conditions with IPTG, together with fitted Gamma distributions against the data (blue lines) and predicted Gamma distributions obtained by using the model values of mean and noise (red lines).
(TIF)

**S5 Fig. Growth curves.** Three different populations (blue, red, and green) were monitored with time. Points correspond to absorbance values, while solid lines come from fitted exponential trends.
(TIF)

**S6 Fig. Relationship between cellular growth rate and volume.** a) Schematics to show that as TC increases, cells grow slower and are bigger. b) Scatter plot between the cube of the forward scattering signal (proxy of cellular volume) and the growth rate for the 81 IPTG and TC conditions (colored by TC condition). An exponential trend was adjusted (solid line).
(TIF)

**S1 Appendix. Stochastic differential equations.** Derivation of the mathematical expressions of noise in eBFP2 and sfGFP having followed a Langevin formalism and the mean-field approximation.
(DOCX)

**S2 Appendix. Gamma distribution.** Derivation of the Gamma distribution for protein expression from a general stochastic differential equation.
(DOCX)

**S1 Data. Flow cytometry data.** Single-cell fluorescence data of eBFP2 and sfGFP for different induction conditions with IPTG and TC after filtering events.
(XLSX)

## Author Contributions

**Conceptualization:** Guillermo Rodrigo.

**Investigation:** Roswitha Dolcemascolo, Lucas Goiriz, Roser Montagud-Martínez, Guillermo Rodrigo.

**Methodology:** Roswitha Dolcemascolo, Lucas Goiriz, Roser Montagud-Martínez, Guillermo Rodrigo.

**Supervision:** Roser Montagud-Martínez, Guillermo Rodrigo.

**Visualization:** Lucas Goiriz, Guillermo Rodrigo.

**Writing – original draft:** Roswitha Dolcemascolo, Lucas Goiriz, Guillermo Rodrigo.

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
