## [Decision Letter · Decision Letter 0]

12 Jul 2021

Dear Dr Rodrigo:

Thank you very much for submitting your manuscript "Gene regulation by a protein translation factor at the single-cell level" (PCOMPBIOL-D-21-00828) for review by PLOS Computational Biology. 

As with all papers reviewed by the journal, your manuscript was reviewed by members of the editorial board and by several independent reviewers. Based on the reviews, we regret that we will not be pursuing this manuscript for publication at PLOS Computational Biology.

Although overall the paper presents new findings, the integration and application of the quantitative analysis is lacking and doesn't follow norms and definitions from previously published work in stochastic gene expression.

The reviews are attached below this email, and we hope you will find them helpful if you decide to revise the manuscript for submission elsewhere. 

While we cannot consider your manuscript further for publication in PLOS Computational Biology, we would like to offer you the option to transfer your submission, with reviews, to PLOS ONE https://www.editorialmanager.com/PONE/

If you DO wish to transfer your submission, please click this link:

<DeepLinkData><DeepLinkTypeID>27</DeepLinkTypeID><peopleID>749731</peopleID><userSecurityID>511488f0-8f5e-4815-a7a9-c79a1db60ee3</userSecurityID><documentID>29736</documentID><revision>0</revision><manuscriptNumber>PCOMPBIOL-D-21-00828</manuscriptNumber><docSecurityID>d43998f3-b36c-436d-8eb1-98fe1a06688d</docSecurityID></DeepLinkData>

If you do NOT wish to transfer your submission, please click this link to decline:

<DeepLinkData><DeepLinkTypeID>28</DeepLinkTypeID><peopleID>749731</peopleID><userSecurityID>511488f0-8f5e-4815-a7a9-c79a1db60ee3</userSecurityID><documentID>29736</documentID><revision>0</revision><manuscriptNumber>PCOMPBIOL-D-21-00828</manuscriptNumber><docSecurityID>d43998f3-b36c-436d-8eb1-98fe1a06688d</docSecurityID></DeepLinkData>

Please note, all PLOS journals are editorially independent and vary in submission requirements.

Should you choose to transfer, your manuscript files, along with the reviewers' comments and their identities will be transferred automatically, and you will receive a confirmation email within 24 hours. Once transferred, your submission will be returned to you so you can check over your record before completing the submission. You may be asked to provide additional information, such as a response to the reviewers' comments. If you have any questions, please contact the editorial office of PLOS ONE https://www.editorialmanager.com/PONE/

We are sorry that the news is not more positive on this occasion, and we hope you will consider PLOS Computational Biology for future submissions. Thank you for your support of PLOS and of open-access publishing.

Sincerely,

David Umulis

Associate Editor

PLOS Computational Biology

Ilya Ioshikhes

Deputy Editor

PLOS Computational Biology

Reviewer's Responses to Questions

**Comments to the Authors: **

Reviewer #1: This manuscript aims to examine the signal transduction and noise-modulating properties of various signaling network structures comprising feedforward loops. All possible network variants are assembled, varying the type of regulation, dual input function and node duplication. The input signal varies in mean but has a constant Fano factor of 1.83. The mean and Fano factor at the output are compared to those at the input. The study identifies the best-performing network structures for signal transduction and noise reduction. 

The manuscript is interesting, timely, well executed and well written. It should be publishable after the following comments are addressed.

(1) The Fano factor measures burstiness (how different a process is from the Poisson process). However, this is not the only measure of noise magnitude. To characterize the standard deviation in terms of the mean, the Coefficient of Variation (CV, defined as standard_deviation/mean) is very widely used. The CV may be preferable for characterizing the quality of signal transduction, whereas the FF is most appropriate to characterize the burstiness of gene expression. For this reason, all the plots showing Fano Factors versus slope should also be made for the CV, using a constant CV at the input.

(2) Keeping the Fano factor constant at the input makes the reader wonder, what would have happened at higher/lower input FF? It would be interesting to explore what happens if the input FF is higher or lower. The same question refers to the input CV. 

(3) The manuscript examines signaling (post-translational) interactions. Similar network structures exist in transcriptional regulatory networks. It would be important to discuss how the results would apply to transcriptional regulatory interactions. A relevant reference for FFL clustering may be PMID:15018656.

(4) “…better noise reduction (lower Fano) correlates with better signal transduction (larger slope).” More specifically, this is true only for slope > 0. However, for slope < 0, better noise reduction (lower Fano) seems to correlate with worse signal transduction. Best signal transduction for slope < 0 is near -1, where the FF is the highest. The same is probably true for coupled motifs.

(5) Besides the FFL, negative feedback can effectively reduce noise, as the authors note. In addition to ref. [13], this is also described in PMID:10850721 and PMID:19279212, which may be worth citing.

Reviewer #2: In this paper Dolcemascolo et al. study noise propagation in a synthetic

genetic network where protein expression is regulated at the level of

translation. The authors used the MS2-MCP system to inhibit the translation

of a super folding GFP (sfGFP) variant and monitored the expression of both

the regulator (MCP fused to enhanced blue fluorescent protein 2 or eBFP2)

and the target (sfGFP) in E. coli. MCP was expressed under the control of a

lac inducible promoter so that the expression level of MCP and hence the

translational repression of sfGFP could be controlled by IPTG treatment.

The authors empirically determined the steady state distributions of eBFP2

and sfGFP at different IPTG concentrations as well as under tetracycline

treatment that modified the growth rate and various kinetic properties of

the cells. The empirical data were analyzed by computing noise defined as

the CV^2, where the CV is the coefficient of variation.

The empirical contributions of this paper are strong: a new synthetic gene

circuit has been established and produces excellent quantitative data at

single-cell resolution. 

The computational/theoretical contributions however are rather weak.

Although the authors call it "stochastic modeling", equations 6 (in

Materials and Methods) do not fit the commonly accepted meaning of the term

in the field of stochastic gene regulation and can best be described as

deterministic curve fitting. A few observations regarding Equations 6:

1. The third term in the expressions for CV^2 in equation 6 performs error

propagation where the standard deviation of a dependent variable is

linearly approximated as a function of the standard deviations of its

independent variables. This is a purely deterministic framework. 

2. The second term in these equations representing "intrinsic" noise

assumes a constant Fano factor. Based on Friedman et al. (PRL 97: 168302),

cited by the authors, and also Thattai and van Oudenaarden (PNAS 98: 8614)

one would expect that the Fano factor is given by b+1 (discrete master

equation) or b (continuous master equation). b is the

average number of proteins produced per burst, which is proportional to the

translation rate. If the translation rate of sfGFP is being regulated by

MCP, the Fano factor (=b) should be dependent on the amount of MCP/eBFP2 and 

not a constant as the authors assume.

3. Based on the above two points, the authors are effectively carrying out

a deterministic curve fitting exercise when modeling the noise.

4. Another problem is the use of the terms "extrinsic" and "intrinsic".

Barring a two-color experiment (two colors of the translationally repressed

protein), it is not possible to decompose the noise into extrinsic and

intrinsic components. Although one may decompose the total noise into two

terms (intrinsic and extrinsic), decomposing further into three or more

terms (each representing averages over subsets of independent variables,

see Swain et al. PNAS 99: 12795) is not possible as far as I can tell. If

the authors have done so, they should include a derivation. Third, the

transfer function noise is also extrinsic since it involves the randomness

of eBFP2 expression but not the inherent stochasticity of translation.

The authors observed a lower level of noise in sfGFP but a clear

mechanistic explanation has not been provided. The assumptions of a gene

transcribed and translated at a constant rate at steady state match the

experimental setup and the Gamma distribution describes the empirical

histograms well. Then according to the results of Friedman et al. (PRL 97:

168302), CV^2 = 1/a (shown as Eq. 10 in Methods), where a is the Gamma

distribution shape parameter that is proportional to the transcription

rate (and not translation rate). This could imply that the lower noise 

in sfGFP is due to higher transcript numbers and not translational 

regulation.

The deterministic Hill equations used for determining the dependence of

mean eBFP2 and sfGFP concentrations on IPTG and eBFP2 concentrations

(Eq. 1) are unusual. The Hill equation for activation that results from

mass kinetics takes the form (X/K)^n/(1 + (X/K)^n) and the Hill equation

for repression takes the form 1/(1 + (X/K)^n). Equation 1 however combines

both repression and activation:

alpha * (sigma + (IPTG/theta)^n)/(1 + (IPTG/theta)^n)

= alpha * [ sigma/(1 + (IPTG/theta)^n) + 

 (IPTG/theta)^n/(1 + (IPTG/theta)^n) ]

According to this equation IPTG both activates and represses eBFP2

simultaneously and sigma is a weight to determine the balance between

activation and repression. This may have been done to achieve 

non-zero eBFP2 expression when there is no IPTG. A basal or leaky level of

eBFP2 expression in the absence of IPTG might be better represented by

adding a constant term to the Hill equation for activation.

In summary, the paper treats propagation of noise in a deterministic

manner and lacks any sort of stochastic treatment of translation noise under

translational regulation.

**Have the authors made all data and (if applicable) computational code underlying the findings in their manuscript fully available?**

Reviewer #1: Yes

Reviewer #2: Yes

PLOS authors have the option to publish the peer review history of their article (what does this mean?). If published, this will include your full peer review and any attached files.

Reviewer #1: No

Reviewer #2: No

---

## [Decision Letter · Decision Letter 1]

17 Dec 2021

Dear Dr Rodrigo,

Thank you very much for submitting your manuscript "Gene regulation by a protein translation factor at the single-cell level" for consideration at PLOS Computational Biology. As with all papers reviewed by the journal, your manuscript was reviewed by members of the editorial board and by several independent reviewers. The reviewers appreciated the attention to an important topic. Based on the reviews, we are likely to accept this manuscript for publication, providing that you modify the manuscript according to the review recommendations.

Sincerely,

David M. Umulis

Associate Editor

PLOS Computational Biology

Ilya Ioshikhes

Deputy Editor

PLOS Computational Biology

[LINK]

Reviewer's Responses to Questions

**Comments to the Authors:**

Reviewer #1: I would like to thank the Authors for responding to the earlier comments. Due to their edits, and due to their responses to both Reviewers’ comments, the manuscript has improved substantially, and it has become more understandable. Now I would like to make some further comments, addressing which could make the manuscript acceptable for publication.

(1) Are there any natural systems what share the architecture of this synthetic system? It would be great to mention a couple of examples. That could increase the relevance of the research.

(2) The manuscript would become much clearer and more readable if the Results is divided into subsections, each with a subtitle. Possible subsections could have subtitles like 1: Constructing the experimental system; 2: Single cell expression measurements; 3: Deterministic and stochastic mathematical modeling; 4: Noise decomposition; 5: Modeling distributions; 6: The effect of translational inhibition.

(3) Is there a native LacI gene in this E. coli strain? The lacI+ feature suggests this. How would native LacI expression affect the system? Likewise, there seems to be a tetR gene in the cells, as suggested by the tetR+ notation in the Methods. If TetR is expressed in the cells, it will bind Tetracycline (TC) and it will prevent translational inhibition by TC. That effect will contribute to the growth rate vs. TC profile in Figure 3C. Does TetR regulate any gene in the cells?

(4) Error bars are not shown in figures, with the justification that the noise (CV^2, standard deviation, etc.) represents the error bars. However, this is not a standard way to omit error bars. Error bars should arise from biological replicates instead of single cell values deviating from the average. Therefore, all the mean and noise curves should have error bars representing the deviations of biological replicates from a mean. In other words, there should be a mean of the mean, a mean of the noise, a standard deviation of the mean and a standard deviation of the noise calculated from at least 3 biological replicates. Likewise, error bars around simulated values could arise from repeated stochastic simulations. See Murphy et al., PNAS 104 (31) 12726-12731 (2007) and Murphy et al., Nucleic Acids Res. 38(8): 2712–2726 (2010).

(5) While the theoretical approach is interesting, it would be helpful to also try some stochastic simulations of the system. Such simulations might predict at least the intrinsic noise behaviors without TC.

(6) Adding TC will generate single-cell differences in cell division rate. The overall growth rate will be the average of all single cell division rates. In other words, there is a fitness average and a fitness noise. In the case of TC treatment, how large is the single-cell fitness noise? These concepts introduced in PLoS Comput. Biol. 8(4): e1002480 (2012) and then revisited in PLoS Comput Biol. 12(3): e1004825 (2016) and PNAS 115 (45) E10797-E10806 (2018) would be worth studying or at least discussing for TC. The fitness noise due to TC could be a source of extrinsic noise that correlates the extrinsic fluctuations of eBFP and sfGFP.

(7) In Equations (6), the constants for extrinsic noise are strong assumptions. Are these constants assumed the same for all IPTG and TC conditions? It may be an issue if TC affects dilution mean & noise.

(8) There are some minor grammar issues that would be helpful to address. For example, in line 51: “pursuit” may be better; line 60: “progress has been made”; in line 119: “MS2CP being” would be better. Also, some clarifications would be useful, such as on line 150: “the total noise of both eBFP and sfGFP”, on line 182: “inverse of the noise (equal to mean^2/variance)”.

Reviewer #2: 1. In response to the criticism that the manuscript appeared not to be

performing stochastic modeling, the authors have clarified that equation 6

has been derived by modeling mRNA/protein concentration dynamics using the

Langevin equation. Expressions for the variance of eBFP2 and sfGFP have

been determined analytically by computing the power spectra and using the

Wiener-Khintchine theorem.

I went back to the original submission and looked again and "Langevin" is

not mentioned even once in the entire manuscript. The results state (lines

115-117 of original ms):

"We then constructed a mathematical model relying on a series of

algebraic equations from basics on the biochemistry of gene expression and

molecular noise propagation [7]"

This sentence can describe almost any model in computational biology. Even

in the Methods section, which should describe the *methods* in sufficient

detail so that the work may be reproduced, the sentence describing the

model states:

"The expressions for the variances in gene expression (or fluorescence

signal) can be derived by following some basic calculations [3,7]."

If the authors do not specify what "basic calculations" were performed, the

readers, including the reviewer, are free to use their imaginations. As it

happens, the same expressions for variance can be derived by error

propagation, with very similar assumptions/approximations (sources of noise

are independent, linear approximation).

The authors have included a derivation of equation 6 in Appendix S1 of the

revised manuscript. The authors still fail to mention/describe the modeling

method or its underlying assumptions, apart from the very generic

descriptions quoted above, in the results or methods. The reader should not

have to refer to other papers or read the appendices to know at a high

level the method used to arrive at equation 6, which is central to the

analysis presented in the paper. This is also important for critically

assessing the manuscript. For example, one crucial assumption is that the

noise sources are independent. This is not strictly true since eBFP2 and

sfGFP share extrinsic noise sources and so the regulation noise from eBFP2

is not independent of extrinsic noise. In this, the manuscript appears to

diverge from the treatment in Pedraza and van Oudenaarden (Science,

307:1965), who modeled transmitted noise as strongly correlated with global

(equivalent to extrinsic in this ms's terminology). These assumptions were not

mentioned at all in the original submission and are only mentioned in the appendix

in the revised version.

The authors should include a short description of the Langevin formalism and its

assumptions in the results and methods sections. They have already

described how they chose the variances for the different noise terms (lines

418-424). A description of the analytical methodology and its assumptions

is missing. It would also be useful to discuss the limitations imposed by

these assumptions in the Discussion section.

2. The lack of sufficient description of the modeling also explains the

confusion regarding the terms "intrinsic" and "extrinsic". The impression

one gets from the manuscript (lines 149-151 of revised ms):

"For that, we first decomposed the total noise into three fundamental

components: extrinsic noise, intrinsic noise, and regulation noise."

is that it is *always* possible to decompose the

noise into three components, which is not correct. Without any further

assumptions, noise can be decomposed into two components (Swain et al. PNAS

99: 12795) but not three. To achieve the decomposition carried out in the

ms, it is necessary to make the additional assumption that "extrinsic",

"intrinsic", and regulatory random variables are independent. A better

wording might be something like:

"We modeled stochastic mRNA/protein expression using Langevin equation.

The rate of change of mRNA concentration was subject to fluctuations from

three sources: .... Assuming that the sources are independent, the

noise can be decomposed into three contributions"

The authors have cited Pedraza and van Oudenaarden (Science,

307:1965), who defined the noise sources more carefully as global,

intrinsic, and plasmid. The authors should consider following that example

since "extrinsic" means all sources that are not inherent to the

transcription/translation process (Elowitz et al., Science 297:1183,

Swain et al. PNAS99: 12795) and includes, among other things, transmitted noise.

3. The authors have satisfactorily addressed all the other points raised in

the review.

4. A general comment: part of the difficulty of understanding the

manuscript has been that it has been written very tersely and is stingy

with explanations, yet it shies away from technical/mathematical

descriptions in the main text. Anything the authors can do to expand

their explanations would help.

For example, it is indeed true that the authors noted that the lower noise

in sfGFP must be due to a high transcription rate (lines 146-148, original

submission), but it is stated in such an off-hand manner:

"This suggested that with a translational control the noise of the regulator is

buffered, as the regulated gene is transcribed at high levels."

and without reference to the theoretical results that it was easily missed.

It would be better to expand this explanation a bit and point out that 1)

the noise is inversely related to the shape parameter of the Gamma

distribution and 2) that the shape parameter is proportional to the

transcription rate (ideally using the mathematical expressions).

**Have the authors made all data and (if applicable) computational code underlying the findings in their manuscript fully available?**

Reviewer #1: Yes

Reviewer #2: Yes

PLOS authors have the option to publish the peer review history of their article (what does this mean?). If published, this will include your full peer review and any attached files.

Reviewer #1: No

Reviewer #2: No

Figure Files:

Data Requirements:

Reproducibility:

References:

---

## [Decision Letter · Decision Letter 2]

7 Apr 2022

Dear Dr Rodrigo,

We are pleased to inform you that your manuscript 'Gene regulation by a protein translation factor at the single-cell level' has been provisionally accepted for publication in PLOS Computational Biology.

Best regards,

David M. Umulis

Associate Editor

PLOS Computational Biology

Ilya Ioshikhes

Deputy Editor

PLOS Computational Biology

Reviewer's Responses to Questions

**Comments to the Authors:**

Reviewer #1: I would like to thank the Authors for addressing my comments. I would like to recommend the publication of the revised manuscript.

Reviewer #2: All my concerns have been addressed.

**Have the authors made all data and (if applicable) computational code underlying the findings in their manuscript fully available?**

Reviewer #1: Yes

Reviewer #2: None

PLOS authors have the option to publish the peer review history of their article (what does this mean?). If published, this will include your full peer review and any attached files.

Reviewer #1: No

Reviewer #2: No

---

## [Editor Report · Acceptance letter]

29 Apr 2022

PCOMPBIOL-D-21-00828R2 

Gene regulation by a protein translation factor at the single-cell level

Dear Dr Rodrigo,

I am pleased to inform you that your manuscript has been formally accepted for publication in PLOS Computational Biology. Your manuscript is now with our production department and you will be notified of the publication date in due course.

With kind regards,

Anita Estes
